# Efficacy of inverted inner limiting membrane flap technique for macular holes of ≤400 μm: A systematic review and meta-analysis

**Seung Min Lee**[1,2], **Ji Woong Lee**[2,3], **Ji Eun Lee**[1,2], **Hee-young Choi**[2,3], **Jong Soo Lee**[2,3], **Iksoo Byon**[2,3]*

**1** Department of Ophthalmology, Research Institute for Convergence of Biomedical Science and Technology, Pusan National University Yangsan Hospital, Yangsan, South Korea, **2** Pusan National University School of Medicine, Yangsan, South Korea, **3** Department of Ophthalmology, Medical Research Institute, Pusan National University Hospital, Busan, South Korea

* isbyon@naver.com, isbyon@pusan.ac.kr

**Data Availability Statement:** All relevant data are within the paper and its Supporting information files.

## Abstract

### Purpose

To evaluate the efficacy of inverted internal limiting membrane (ILM) flap technique in full-thickness macular holes (MHs) with a size of ≤400 μm compared to the ILM peeling technique.

### Methods

Related literatures that compared inverted ILM flap and ILM peeling in MHs ≤ 400 μm were reviewed by searching electronic databases including Pubmed, EMbase, ClinicalTrials.gov, and Cochrane Library up to April 2023. The primary outcome measure was hole closure rate, and the secondary outcome measures were the mean postoperative best-corrected visual acuity (BCVA), retinal sensitivity, and outer status of the retinal layers, including the external limiting membrane and ellipsoid zone. The quality of the articles was assessed according to the revised version of the Cochrane risk-of-bias tool for randomized trials or the Newcastle–Ottawa scale. In the case of heterogeneity, a sensitivity analysis was conducted, and publication bias was visually evaluated using a funnel plot.

### Results

This review included six studies with 610 eyes for the primary outcome and 385 eyes for the secondary outcomes, which were two randomized control trials and four retrospective studies. Pooled data revealed that the overall MH closure rate was 99.4% in the inverted ILM flap group and 96.2% in the ILM peeling group, without significant difference between the two groups (odds ratio = 3.91; 95% confidence interval, 0.82~18.69; P = 0.09). The inverted ILM flap technique did not have a favorable effect on the BCVA, retinal sensitivity, or recovery of the outer retinal layers. These results were consistent with those of the subgroup analysis of the different follow-up periods. No significant publication bias was observed.

**Funding:** The author(s) received no specific funding for this work.

**Competing interests:** The authors have declared that no competing interests exist.

## Conclusion

In eyes with MHs of ≤400 µm, both techniques demonstrated excellent surgical outcomes without significant differences. Therefore, surgical techniques can be selected according to surgeon preferences.

## Introduction

A macular hole (MH) is a full-thickness retinal defect of the fovea, and surgical repair requires improvements in the foveal structure and visual acuity [1]. Since Eckardt et al. introduced the internal limiting membrane (ILM) peeling (ILMP) technique, which leads to successful hole closure of approximately 90% in primary surgery [2–4]. However, such a high success rate is not achieved in chronic large MHs (>400 µm) [5–7]. Therefore, various techniques, including autologous concentrated platelets and autologous serum, have been applied, although they are not widely accepted owing to conflicting outcomes in the treatment of large MHs [8,9].

In 2010, Michalewska et al. reported an innovative surgical technique using an inverted ILM flap (ILMF) with a high closure rate for large MHs [5]. Many clinical studies and meta-analyses have confirmed that the ILMF technique is superior to the ILMP technique for hole closure and visual recovery in complicated large MHs, including myopic MH and MH retinal detachment [6,7,10,11]. However, considering the surgical procedure, the ILMF technique requires more effort to stabilize, which is a potential difficulty in performing this technique. In cases of inserted ILMF inside the hole, visual and anatomical recovery can be limited [12–14]

Regarding excellent surgical outcomes of ILMP for the treatment of small- and medium-sized MHs (≤400 µm), selecting between ILMP and ILMF as the primary surgical procedure is controversial [14–16]. In addition, a paucity of literature, insufficient sample sizes, and inadequate levels of evidence for comparison studies between the two surgical procedures limit our understanding [17–22].

Therefore, we conducted a meta-analysis of all available randomized control trials and other comparative studies to evaluate the efficacy of the ILMF technique compared to the ILMP technique for MHs of ≤400 µm in diameter.

## Materials and methods

### Search strategy

We comprehensively searched for relevant literature in electronic databases, including PubMed, Embase, ClinicalTrials.gov, and Cochrane Library until April 10, 2023. The terms used for systematic search were as follows: ("macular hole" OR "MH") AND ("inverted flap" OR "Internal limiting membrane flap" OR "inner limiting membrane flap" OR "ILM flap") AND ("small" OR "medium"). This meta-analysis was conducted according to the Cochrane Handbook for Systematic Reviews of Interventions version 6.3 and the Preferred Reporting Items for Systematic Reviews and Meta-Analysis (PRISMA) Statement [23,24, S1 Checklist]. No restrictions on language and publication year were applied when searching the electronic databases. The search results were imported into the reference management software (End-Note 20, Thomson Reuters, New York, NY, USA), and duplicate titles were removed from the merged list of literature. The titles and abstracts were checked to exclude irrelevant studies, and the full texts of the relevant literature were checked to confirm whether they satisfied the inclusion and exclusion criteria. In addition to the reviewed articles, their reference lists were explored as additional eligible sources.

## Inclusion and exclusion criteria

We used the following inclusion criteria according to population, intervention, comparison, outcomes, and study design protocol.

First, for population (participants), patients diagnosed with idiopathic full-thickness MH and a minimum linear diameter of the MH ≤ 400 μm according to the International Vitreo-macular Traction Study Group classification and followed up for >1 month after the procedure were included in the present study [25].

Second, for intervention, ILMF with the ILMP technique in the experimental group and the ILMP-only technique in the control group were compared. No limitations on the flab stabilizer, gauge of the vitrectomy system, and combination of phacoemulsification or tamponade except silicone oil were applied.

Third, for outcome measures, the primary outcome measure was anatomical hole closure rate in the first operation. Type 1 MH closure should be confirmed using optical coherence tomography (OCT) described by Kang et al. [26]. The secondary outcome measures were visual function and recovery of the outer retinal layers. Visual function was evaluated using the best-corrected visual acuity (BCVA) and retinal sensitivity. BCVA was converted to the logarithm of the minimum angle of resolution. The BCVA before surgery and 3, 6, and 12 months after surgery were compared. Retinal sensitivity measured by microperimetry in the central 8°–10° was used for the analysis. Recovery of the outer retinal layers was defined as a continuous line of the external limiting membrane (ELM) or ellipsoid zone as confirmed using OCT. One or more outcomes were reported in each selected study.

Lastly, prospective randomized controlled trials (RCTs), prospective non-RCT studies, and retrospective comparative studies were included.

The exclusion criteria were as follows: MHs with complications, including severe cataracts hindering retinal measurement, glaucoma, degenerative myopia, retinal detachment, ocular inflammation secondary to vitrectomy, and trauma; data required for this meta-analysis were not provided or not available to yield outcomes; poor-quality literature; duplicate publication using the same materials; and care reports, case series without comparative data, surgical techniques, or review articles.

## Data extraction

The literature was reviewed, and data were independently extracted by two investigators (S.M. L and I.B.). Any discrepancies in the literature selection and data extraction were resolved via discussion with a third researcher or consultation with an expert. The extracted data included the following: first author, year of publication, country, study design, sample size of each group, mean age, MH size, MH closure rate, preoperative BCVA, postoperative BCVA at 3, 6, and 12 months, retinal sensitivity, and recovery rate of the EZ and ELM.

## Qualitative assessment

The quality of the selected RCT was evaluated using the revised version of the Cochrane risk of bias tool for randomized trials (RoB2) [24]. Bias was evaluated by subdividing it into five domains: (1) the randomization process, (2) deviations from intended interventions, (3) missing outcome data, (4) outcome measurement, and (5) selection of the reported result. Each domain was assigned a "low risk of bias," "some concerns," or "high risk of bias." The overall results for each domain are presented according to the RoB2 process.

The methodological quality of non-randomized studies, including prospective and retrospective case–control studies, was assessed using the Newcastle–Ottawa scale (NOS) [27]. Each study rated in three categories including "selection," "comparability" and "exposure." A study

can award one star for each item in selection and exposure, and earn up to two stars for comparability. In this study, the quality of the study was defined as low, moderate, and high according to the NOS scores of 0–3, 4–6, and 7–9, respectively. All items were independently evaluated by two researchers (S.M.L and I.B.), and assessment discordance was resolved through discussion or expert consultation.

## Statistical analysis

Data were statistically analyzed using Review Manager 5.30 (Cochrane Collaboration, Oxford, UK). To analyze dichotomous variables, including rate of MH closure and recovery of ELM and EZ, the Mantel–Haenszel (M–H) method was used, and results were presented as odds ratios (OR) and 95% confidence intervals (CI). For analysis of continuous variables, including BCVA and retinal sensitivity, the weighted mean difference (WMD) with 95% CI was calculated using the inverse-variance approach. Statistical significance was defined as $P < 0.05$. Significant heterogeneity was defined characterized by a statistically significant result of the $\chi^2$ test ($P < 0.1$) or a >50% result in Higgin's $I^2$ statics [28]. A random-effects model was used for cases with significant heterogeneity, whereas a fixed-effects model was used when no significant heterogeneity is observed [29]. To verify the robustness and credibility of the results in this meta-analysis, a sensitivity analysis was conducted by removing the included studies individually until the source of heterogeneity was identified, resulting in resolving significant heterogeneity after removal. Subgroup analysis was also used to perform a sensitivity analysis. Owing to the limited number of included studies, evaluation of publication biases by subgroup analysis and asymmetry assessment of the funnel plot were not conducted, and funnel plots were presented to visualize the evaluation for publication bias.

We merged data that were originally divided into subgroups in the form of mean values and standard deviations in accordance with the formula in Chapter 6 of the Cochrane Handbook [24]. When the standard deviation was not presented, we used the formula in the Cochrane Handbook and the method of Wan et al. to obtain the standard deviation from the range or interquartile ranges [24,30].

## Results

### Study selection

In total, 356 studies were identified through the database searches. Among these, 82 duplicates were removed, and 262 records of the remaining 274 articles were excluded after a preliminary review by checking the titles and abstracts for irrelevant topics, non-comparative studies, reviews, surgical techniques, and duplicated results of RCTs due to publication. A total of 12 studies were retrieved for full-text review, although 2 studies were discarded, as one RCT had no results, and the other had no full text available. Among the remaining 10 eligible studies, four studies, including one article with different surgical techniques and three articles without MH size categorization, were excluded after reading and evaluating the full text. Six studies met the inclusion criteria. Two studies were RCTs, and four were retrospective case–control studies [17–22]. Fig 1 illustrates the study selection process.

### Quality assessment

The quality of the two RCTs was assessed in accordance with the ROB2 and is summarized in Fig 2 [25,26].

Both the studies had a low overall risk of selection bias. In the retrospective case–control studies, four studies were evaluated using the NOS scale, and all studies fulfilled the high-

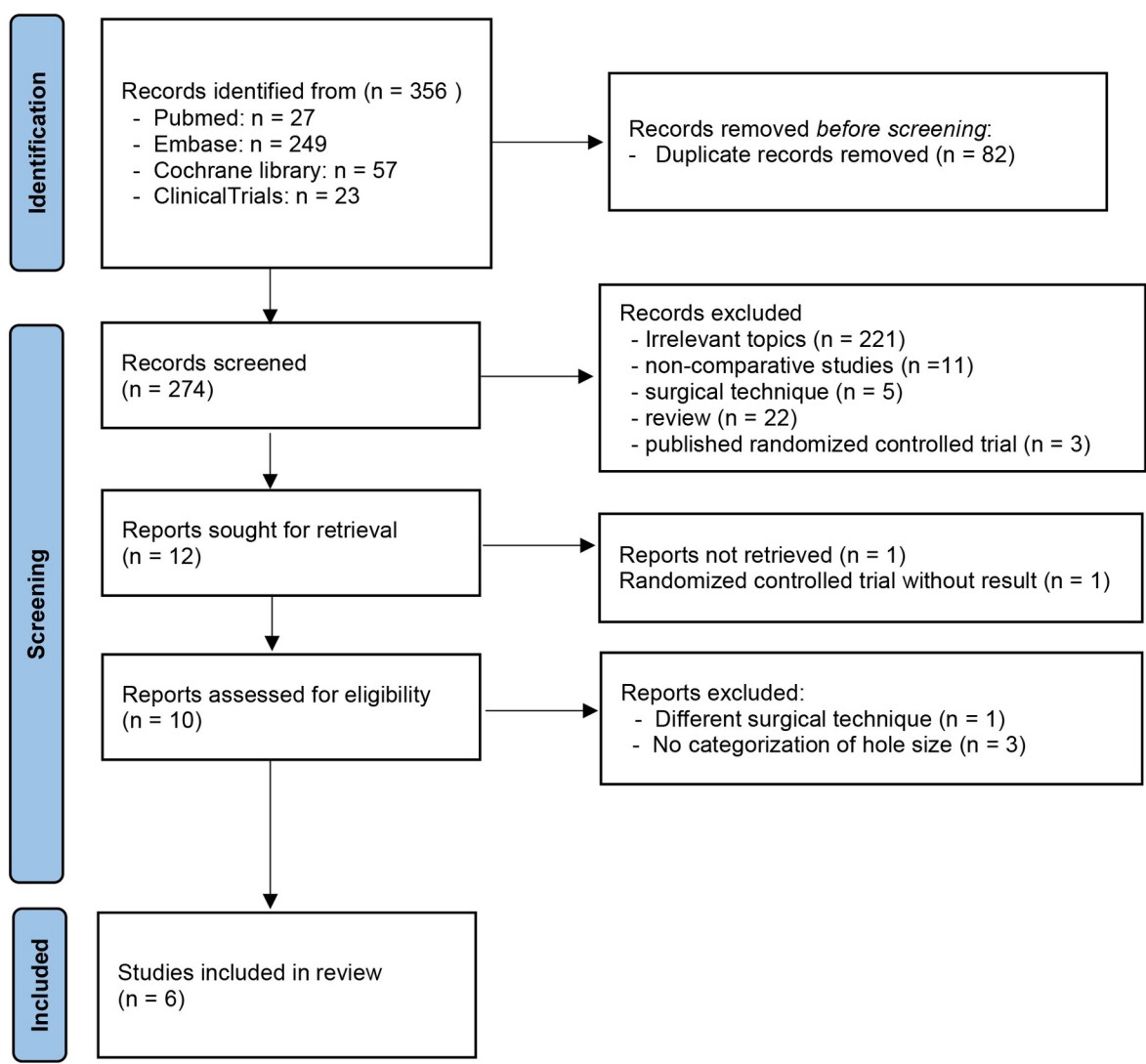

**Fig 1. Flow diagram of the literature screening process according to the PRISMA-p guideline.**

quality criteria with overall 7 to 9 stars [17–19,22] The qualitative assessment of the case–control studies is presented in Table 1.

## Study characteristics

These studies were published between 2021 and 2023. Two studies were conducted in Italy [19,21], one in Japan [22], one in Taiwan [18], one in Austria [20], and one each in Germany and the United Kingdom [17]. The characteristics of the included studies are summarized in Table 2.

Since Chou et al. documented additional data for the MH closure rate [17], the number of eyes included in this meta-analysis was 610 for the MH closure rate analysis and 385 for the secondary outcomes. In total, 178 eyes in the ILMF group and 432 eyes in the ILMP group for MH closure and 182 eyes in the ILMF group and 203 eyes in the ILMP group for the secondary outcomes were analyzed. The sample sizes of the included studies ranged from 16 to 117. The

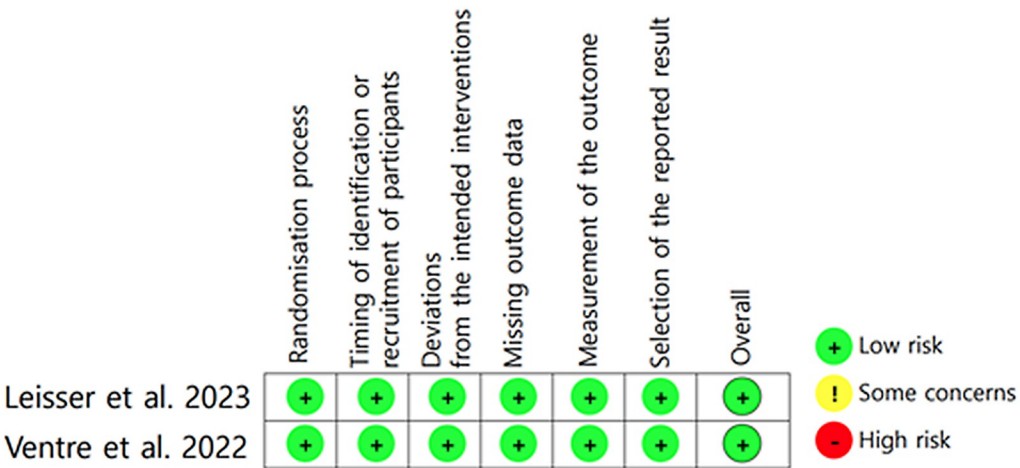

**Fig 2. Assessment of the risk of bias in included randomized controlled trials according to the revised version of Cochrane risk-of-bias tool for randomized trials.** In two studies, overall assessment was identified as low risk.

mean age of the patients ranged from 58 to 71 years. The vital dyes used for ILM staining were brilliant blue G and/or trypan blue in three studies [17,18,21], blue life and trypan blue in one study [19], indocyanine green in one study [18], and unknown in one study [22]. Gas tamponade was used at the end of the surgery in all studies, and the face-down posture was maintained for 1–7 days. The follow-up duration was >1 month in one study [18], 3 months in one study [20], 6 months in one study [19], and 12 months in three studies [17,21,22].

## MH closure

MH closure (primary outcome measure) was achieved in 177 of 178 eyes (99.4%) in the ILMF group and in 407 of 423 (96.2%) eyes in the ILMP group. In the study by Iuliano et al., as unclosed MH were not included, they were excluded from MH closure rate assessment. The combined result of five studies revealed no significant difference in MH closure rate between

**Table 1. Quality assessment using the Newcastle–Ottawa scale for non-randomized controlled trial studies.**

| Author | Selection | | | | Comparability | Outcome | | | Total score |
|---|---|---|---|---|---|---|---|---|---|
| | 1 | 2 | 3 | 4 | | 1 | 2 | 3 | |
| Baumann et al. 2021 [17] | ☆ | ☆ | ☆ | ☆ | ☆☆ | ☆ | ☆ | ☆ | 9 |
| Chou et al. 2021 [18] | ☆ | ☆ | | ☆ | ☆☆ | ☆ | ☆ | ☆ | 8 |
| Iuliano et al. 2023 [19] | ☆ | | | ☆ | ☆☆ | ☆ | ☆ | ☆ | 7 |
| Yamada et al. 2022 [22] | ☆ | ☆ | | ☆ | ☆☆ | ☆ | ☆ | ☆ | 8 |

Selection 1: Adequate case definition.

Selection 2: Representativeness of the cases.

Selection 3: Selection of controls.

Selection 4: Definition of controls.

Comparability 1: Comparison of cases and controls based on the design or analysis.

Outcome 1: Assessment of exposure.

Outcome 2: Same method of ascertainment for cases and controls.

Outcome 3: Nonresponse rate.

**Table 2. Characteristics of studies included in the meta-analysis: Baseline characteristics of the inverted internal limiting membrane flap and internal limiting membrane peeling groups.**

| First author, Year | Study design | Country | Tamponade | Dye | Flap insertion | Group | No. of eyes | Mean age (years) (SD) | Baseline MH size(µm) | Hole closure rate | Preoperative BCVA (logMAR) | Postoperative BCVA (logMAR) | Follow-up time (months) |
|---|---|---|---|---|---|---|---|---|---|---|---|---|---|
| Baumann et al., [17] 2021 | Non-RCT | Germany, UK 2017–2019 | SF6, C2F2, or C3F8 | BBG ± TP | Covering | ILMF | 24 | 63.1 ± 7.7 | 282 ± 104 | 100% (68/68) | 0.77 ± 0.32 | 0.18 ± 0.12 | 12 |
| | | | | | | ILMP | 36 | 70.5 ± 8.2 | 236 ± 86 | 96% (296/308) | 0.74 ± 0.30 | 0.26 ± 0.20 | |
| Chou et al.,[18] 2021 | Non-RCT | Taiwan 2012–2020 | SF6 or C3F8 | ICG | Covering | ILMF | 55 | 61.3 ± 6.4 | 251.9 ± 76.7 | 98% (54/55) | 1.05 ± 0.43 | 0.48 ± 0.33 | 1~12 |
| | | | | | | ILMP | 62 | 58.8 ± 9.9 | 261.6 ± 99.7 | 97% (60/62) | 1.05 ± 0.36 | 0.51 ± 0.36 | |
| Iuliano et al.,[19] 2023 | Non-RCT | Italy 2014–2021 | SF6 | Bluelife and TP | Covering | ILMF | 50 | 69.8 ± 11.3 | 237.9 ± 39.2 | NA | 0.67 ± 0.13 | 0.16 ± 0.11 | 6 |
| | | | | | | ILMP | 50 | 70.6 ± 10.1 | 242.0 ± 33.9 | NA | 0.68 ± 0.15 | 0.12 ± 0.10 | |
| Lesisser et al.,[20] 2023 | RCT | Austria 2018–2021 | SF6 | BBG and TP | Covering | ILMF | 7 | 71 ± 7 | 275 ± 90 | 100% (7/7) | 0.70 ± 0.11 | 0.30 ± 0.14 | 3 |
| | | | | | | ILMP | 9 | 67 ± 5 | 244 ± 101 | 100% (9/9) | 0.76 ± 0.26 | 0.34 ± 0.10 | |
| Ventre et al.,[21] 2022 | RCT | Italy 2020 | SF6 | BBG and TP | Covering | ILMF | 25 | 62 ± 5 | 269 ± 52 | 100% (25/25) | 0.76 ± 0.22 | 0.22 ± 0.11 | 12 |
| | | | | | | ILMP | 25 | 64 ± 5 | 254 ± 70 | 100% (25/25) | 0.72 ± 0.21 | 0.19 ± 0.14 | |
| Yamada et al.,[22] 2022 | Non-RCT | Japan 2014–2017 | SF6 | NA | Covering or filling | ILMF | 21 | 66.2 ± 10.6 | 278.6 ± 80.7 | 100% (21/21) | 0.71 ± 0.3 | 0.28 ± 0.24 | 12 |
| | | | | | | ILMP | 21 | 66.6 ± 7.0 | 276.0 ± 84.5 | 90% (19/21) | 0.73 ± 0.3 | 0.24 ± 0.24 | |

Abbreviations: BBG, brilliant blue G; BCVA, best corrected visual acuity; C2F2, perfluoroethane; C3F8, perfluoropropane; FU, follow-up; ICG, indocyanine green; ILMF, inverted internal limiting membrane flap; ILMP, internal limiting membrane peeling; RCT, randomized controlled trial; logMAR, logarithm of the minimum angle of resolution; MH, macular hole; NA, not available for statistics; No, number; SD, standard deviation; SF6, sulfur hexafluoride; TP, Trypan blue; UK, United Kingdom.

the ILMF and ILMP groups in OR comparison (OR = 3.91; 95% CI, 0.82 to 18.69; P = 0.09), with no heterogeneity (P = 0.77 in $\chi^2$ test, $I^2$ = 0%) (Fig 3).

In order to exclude the effect of the additional data for MH closure described by Chou et al., when the MH closure rate of the two groups was compared without result of Chou et al., no significant difference in the results was observed (OR = 5.68; 95% CI, 0.67~47.91; P = 0.11).

## Visual acuity recovery

Data from six studies were used to assess the preoperative and postoperative BCVA. The preoperative BCVA was 0.72 ± 0.30 with the ILMF group and 0.74 ± 0.28 with the ILMP group. No significant difference in baseline BCVA was observed between the ILMF and ILMP groups (WMD, 0.00; 95% CI, −0.04 to 0.05; P = 0.89) (Fig 4).

The postoperative BCVA of six studies were used to compare the differences between the ILMF and ILMP groups. The postoperative BCVA was 0.20 ± 0.20 in the ILMF group and 0.18 ± 0.24 in the ILMP group, without significant difference between the two groups (WMD, 0.01; 95% CI, −0.02 to 0.04; P = 0.37). Because of no substantial heterogeneity (P = 0.15, $I^2$ = 38%), a fixed-effects model was used. A sensitivity analysis was performed, and heterogeneity

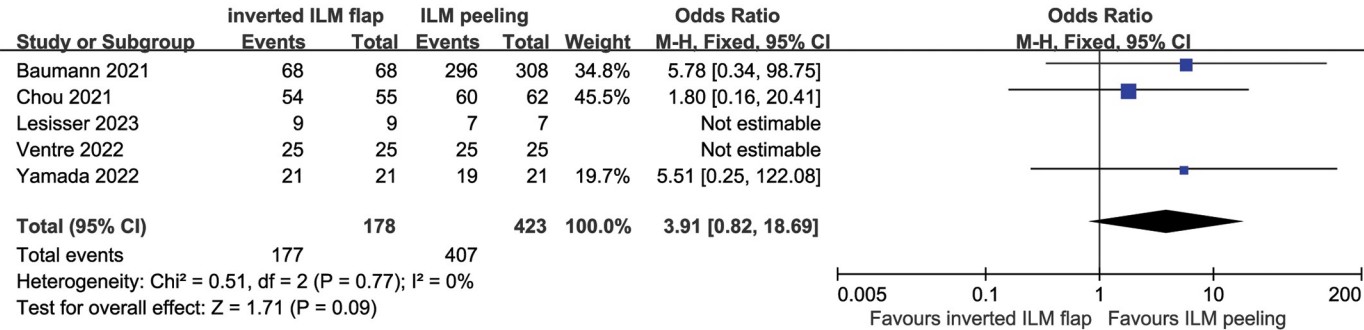

**Fig 3. Forest plot of comparison of macular hole closure rate between the inverted internal limiting membrane flap and internal limiting membrane peeling groups.** No significant difference was observed between two groups. A fixed-effects model was used with no heterogeneity. CL, confidence interval; ILM, internal limiting membrane; M–H, Mantel–Haenszel.

was eliminated after removing the result of Baumann et al. However, it did not alter the result (WMD, 0.03; 95% CI, −0.00 to 0.06; P = 0.08).

A subgroup analysis based on follow-up duration was performed to eliminate the effects of different follow-up times. The data for postoperative BCVA at 3, 6, and 12 months were available in three, three, and four articles, respectively. The meta-analysis demonstrated no significant benefits of the ILMF technique over the ILMP technique in visual recovery over 12 months of observation (Fig 4). The WMD (95% CI) of BCVA between both groups at 3, 6, and 12 months were −0.06 (−0.24 to 0.12), 0.00 (−0.11 to 0.12), and −0.01 (−0.06 to 0.03), respectively. Owing to heterogeneity of pooled data at 3 months and 6 months (P = 0.008; $I^2$ = 79%, P = 0.004; $I^2$ = 70%), a random-effects model was used to analyze the 3 months and 6 months results.

## Retinal sensitivity

Two studies which conducted retinal sensitivity test and compared retinal sensitivity between the ILMF and ILMP groups identified no significant difference in preoperative and postoperative values between the two groups without heterogeneity (WMD, 0.29 and −0.81; 95% CI, −0.46–1.04 and −2.14–0.51; P = 0.45 and 0.23) (Fig 5).

Since postoperative examinations of retinal sensitivity were performed at different times, at 6 months and 12 months postoperatively, respectively, in the two studies, and heterogeneity was observed, a random-effects model was used (P = 0.06, $I^2$ = 71%). A sensitivity analysis could not be performed because only two studies were related.

## Outer retinal anatomical recovery

The postoperative recovery of ELM was compared between the ILMF and ILMP groups in four studies. Complete recovery of ELM did not differ between two groups (WMD, 0.83; 95% CI, 0.44–1.55; P = 0.56), and no significant heterogeneity was observed. When the subgroup analysis was performed regarding the follow-up period, ELM recovery was observed in 82% (range: 62%–93%) and 77% (range: 75%–80%) at 3 months, 61% (range: 42%–95%) and 68% (range: 50%–88%) at 6 months, and 92% (range: 72%–100%) and 91% (range: 85%–96%) at 12 months in the ILMF group and ILMP groups, respectively. No significant difference in the meta-analysis was observed between the two groups (Fig 6).

As heterogeneity was observed in the 3-month and 12-month data, a random-effects model was used (P = 0.03, $I^2$ = 70%; P = 0.12, $I^2$ = 53%). Because the available data were from one

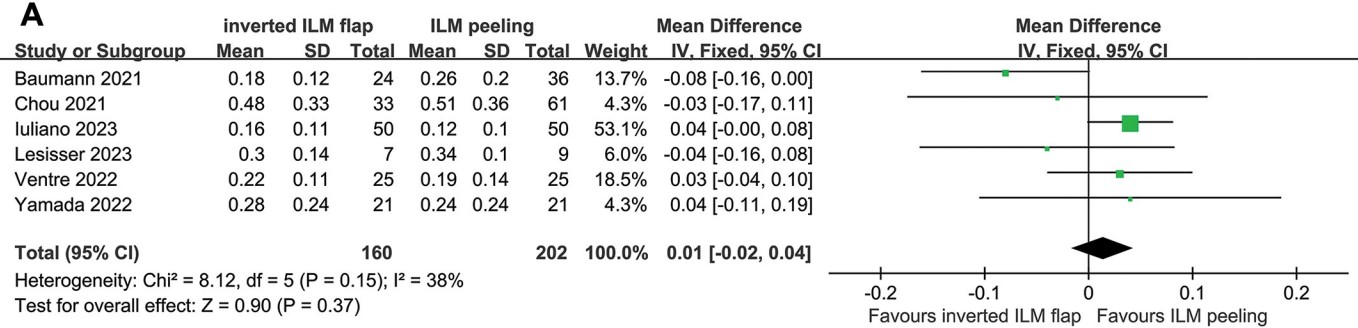

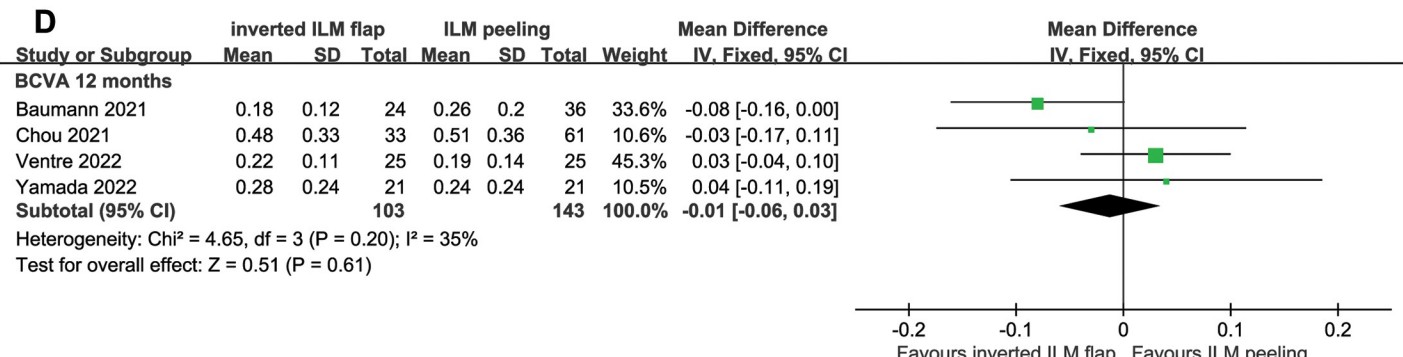

**Fig 4. Forest plots of comparison of postoperative best-corrected visual acuity (BCVA) between the inverted internal limiting membrane flap and internal limiting membrane peeling groups.** (A) Forest plot of postoperative BCVA and (B)–(D) forest plots of postoperative BCVA based on the follow-up duration as the subgroup analysis. The subgroup analysis compared BCVA between two groups at (B) 3 months, (C) 6 months, and (D) 12 months postoperatively, respectively, and no differences were identified. A random-effects model was used in the analysis at 3 months and 6 months since significant heterogeneity was observed. CL, confidence interval; ILM, internal limiting membrane; IV, inverse variance; SD, standard deviation.

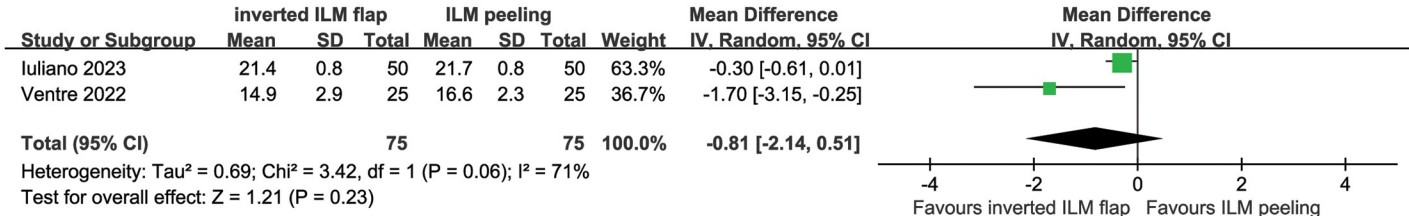

**Fig 5. Forest plots of comparison of retinal sensitivity between the inverted internal limiting membrane flap and internal limiting membrane peeling groups.** No significant difference was observed between two groups. Owing to significant heterogeneity, a random-effects model was used. CL, confidence interval; ILM, internal limiting membrane; IV, inverse variance; SD, standard deviation.

study with 3-month and 12-month data, one study with only 6-month data, and two studies with all follow-up data in the subgroup analysis, the number of studies in the pooled data for each follow-up period was three.

EZ was completely recovered in 28% (range, 9%–43%) and 27% (range, 8%–48%) at 3 months and 47% (range, 40%–52%) and 52% (range, 44%–64%) at 6 months in the ILMF and ILMP groups, respectively. No significant difference in postoperative EZ recovery was observed between the two groups without heterogeneity (Fig 7).

Available data were obtained from one study with 3-month data, one study with 6-month data, and two articles with follow-up data. In the subgroup analysis, three studies were included in the pooled data for each follow-up period. Recovery of the EZ status at 12 months was not analyzed because of the small number of patients included.

## Publication bias analysis

Funnel plots for publication bias assessment of the MH closure rate, preoperative and postoperative BCVA, and ELM recovery are presented in Fig 8.

Data on the MH closure rate in the included studies revealed scattered points distributed in the middle of the inverted funnel. In the pre- and postoperative BCVA, scattered points were distributed near the mean value of the mean difference. In the funnel plot of the ELM recovery rate, positions for studies with small sample sizes or lower mean values appeared vacant. Because all studies were located within the inverted funnel range except for the study by Baumann et al., in which postoperative BCVA was located slightly outside the funnel shape, publication bias in postoperative BCVA was a concern. However, because the results of the sensitivity analysis were not significantly different, and no evidence of publication bias for the remaining items was noted, the publication bias may be low, and the conclusion is relatively reliable.

## Discussion

In the present meta-analysis, six literatures including two RCTs with low risk-of-bias and four retrospective studies with high-quality comparing ILMF technique and ILMP technique for MH of <400 μm were reviewed. A total of 610 eyes were included in the comparison of MH closure rate (178 eyes in the ILMF group and 432 eyes in the ILMP group), and 385 eyes were included in the secondary outcome measures (182 eyes in the ILMF group and 203 eyes in the ILMP group). In the results of the primary outcome measure, the MH closure rate, the analysis of integrated data revealed no more favorable results of the ILMF technique over ILMP technique for MH closure, as both techniques demonstrated excellent success rates of 99.4% in the ILMF group and 96.2% in the ILMP group, respectively. For secondary outcome measures,

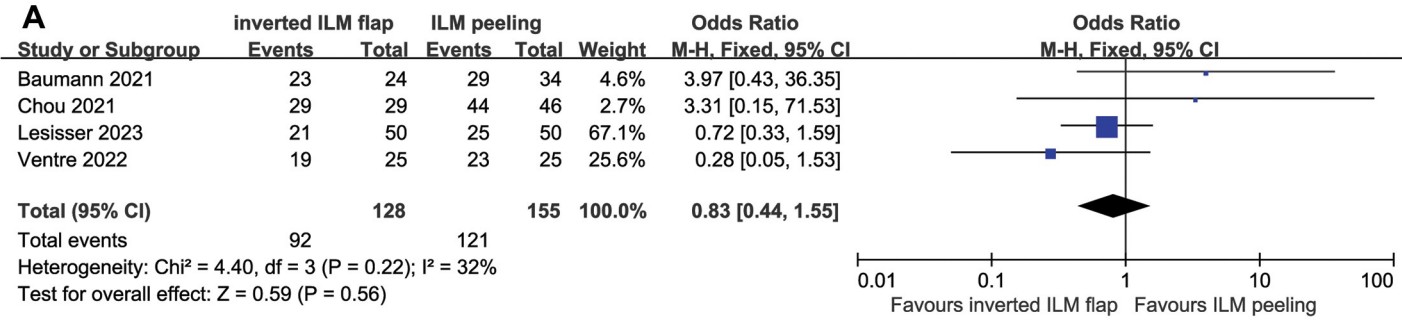

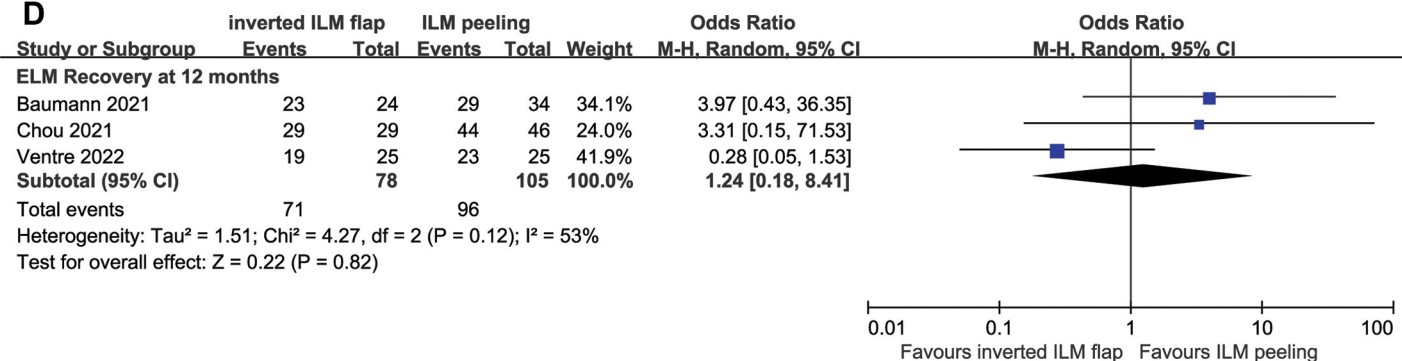

**Fig 6. Forest plots of comparison of complete recovery rate of external limiting membrane between the inverted internal limiting membrane flap and internal limiting membrane peeling groups.** (A) Comparison of external limiting membrane recovery after surgery and (B)–(D) the result of subgroup analysis based on follow-up duration. The follow-up duration was subdivided into (B) 3 months, (C) 6 months, and (D) 12 months after surgery, and no significant difference at each period was observed between the two groups. A random-effects model was used in the analysis in the 3-month and 12-month results since a significant heterogeneity was observed. CL, confidence interval; ILM, internal limiting membrane; M–H, Mantel–Haenszel.

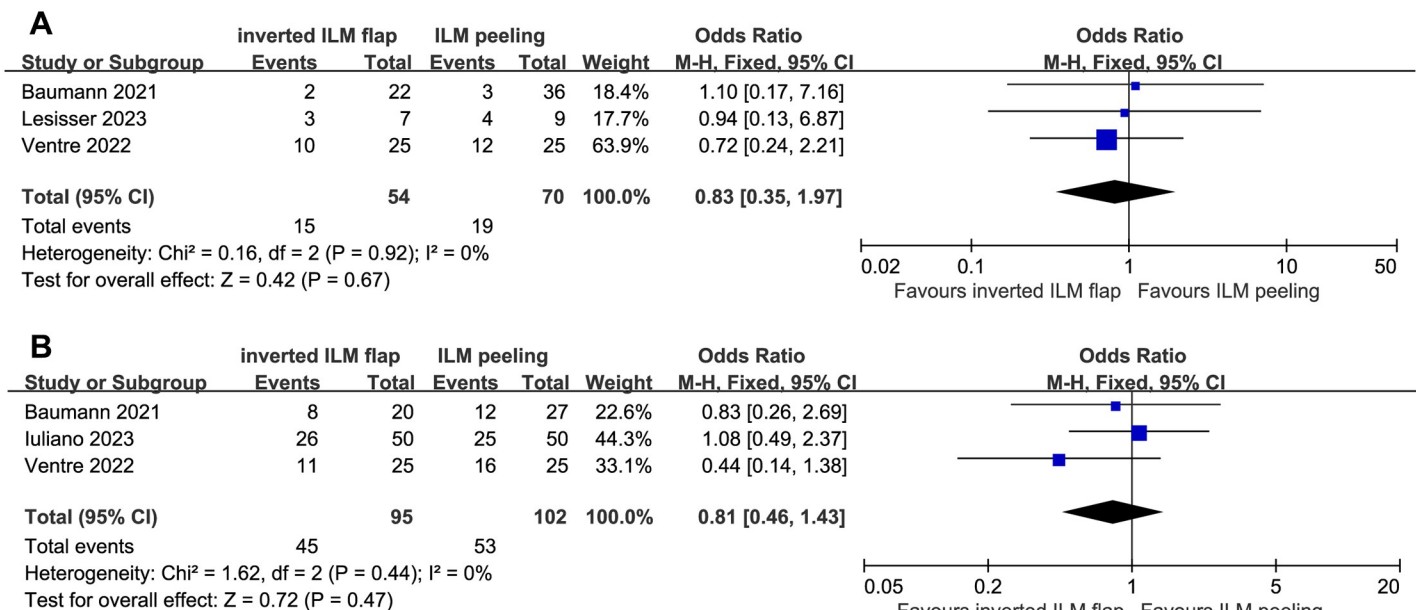

**Fig 7. Forest plots of comparison of complete recovery rate of ellipsoid zone between the inverted internal limiting membrane flap and internal limiting membrane peeling groups.** Comparison of ellipsoidal zone recovery at (A) 3 months and (B) 6 months after surgery revealed no significant difference between the two groups. CL, confidence interval; ILM, internal limiting membrane; M–H, Mantel–Haenszel.

including the recovery of visual acuity, retinal sensitivity, and recovery of the EZ and ELM, no significant differences were observed between the two groups. No favorable technique was identified in the subgroup analyses for the recovery of visual acuity and outer retinal structure at each follow-up period.

Considering that the main cause of idiopathic MH is anteroposterior and tangential traction, vitrectomy to release vitreous traction and gas tamponade is effective for hole closure. The success rate of surgery has been reported as 69%–78% [1,8,31,32]. Additionally, the ILMP technique was introduced to achieve complete traction removal around the hole. It increases the hole closure rate up to approximately ≥90% by reducing of retinal stiffness, removing residual vitreous cortex, and stimulating retinal glial proliferation after ILM removal [31,33–36]. Despite the introduction of the ILMP technique, refractory MHs, including large and myopic MH, present with low closure rates, even when additional procedures are applied [5,37,38]. ILMF techniques, initially introduced for large MH, have demonstrated high closure rates in various refractory MH surgeries [5,6,37,39]. Meta-analyses have demonstrated that the ILMF technique has a significant advantage over the ILMP technique for large MH [10,40]. The success rate can be explained as follows. The ILM flap acts as a scaffold for glial cell proliferation, a source of neurotrophic and growth factors, and a barrier to turbulent flow by covering the hole [39,41,42].

Regarding the high closing rate of large MH, the ILMF technique has been introduced for small- to medium-sized MHs. However, whether ILMF is superior to ILMP, particularly in terms of visual outcomes, is unclear [11,15–22]. In terms of surgical preference, the ILMP technique is relatively easier and simpler than the ILMF technique for stabilizing the flap over the hole. The ILMF technique requires more surgical time and skill to create and stabilize the flap, which can be a potential hurdle for beginners to perform this technique. In addition, excessive glial proliferation by the ILM flap is possible following hole closure, which can limit the

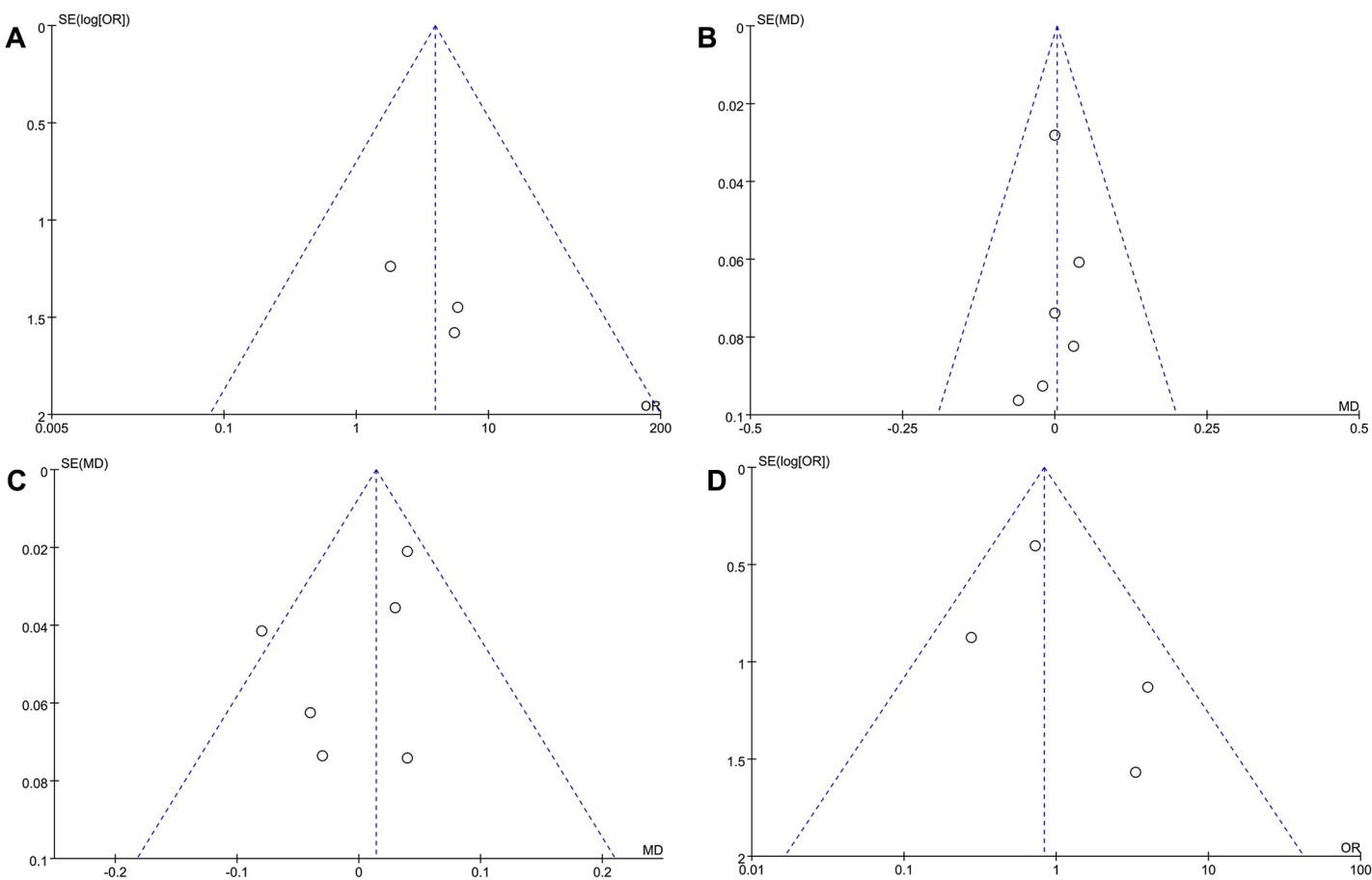

**Fig 8. Funnel plot for evaluating the publication bias.** (A) Macular hole closure rate; (B) preoperative best-corrected visual acuity (BCVA); (C) postoperative BCVA; (D) complete recovery rate of external limiting membrane. (A)–(D) Scattered points corresponding to the included studies were mostly distributed within the range of inverted funnel. (C) One point of study in postoperative BCVA was located slightly out of the inverted funnel. A relatively low publication bias could have influenced the results. OR, odds ratio; MD, mean difference; SE, standard error.

recovery of visual acuity and the outer retinal layer [12–14]. The ILMF technique has the following advantages. The foveal contour can improve with time after the disappearance of the gas tamponade because the ILM flap over the hole acts as a lid to protect the hole and promote the recovery of retinal tissues [7,18]. In this regard, the more suitable primary surgical technique in MHs >400 μm is debatable [14–16]. However, our understanding of this issue is limited because of the paucity of literature and conflicting results from a few studies comparing ILMF and ILMP techniques [17–22].

Five studies were included in analyzing the closure rate in MHs <400 μm in this meta-analysis [17,18,20–22]. No significant differences between the two surgical techniques. This finding was confirmed in a meta-analysis of 610 eyes. This indicated that the ILMF technique does not provide additional benefits in anatomical success of MHs of <400 μm. This can be explained as follows. ILMP have demonstrated very high closure rates ranging from 96% to 100%. Therefore, demonstrating the superiority of the ILMF technique in closing holes in small-to medium-sized MHs was difficult [15–17]. MHs with a mean diameter of 312.5 ± 105.2 μm were reportedly closed 24 h postoperatively [43]. MH closure was also observed in all eyes when the sum length of both hole slopes was >90% of the base diameter of the hole, potentially

with most of the holes covered by tissue [44]. This implies that tissue apposition around the hole occurs more easily and rapidly after surgery. Therefore, an ILM flap covering the holes might not be needed in small-to-medium-sized MHs.

To compare outer retinal integrity, four studies were included in the ELM analysis and three studies were included in the EZ analysis. No significant differences in outer retinal recovery were observed in our meta-analysis. In terms of recovery time, no significant difference was observed between the ELM and EZ at each follow-up visit. Two previous studies demonstrated favorable outcomes with the ILMP technique [19,22]. However, contradictory outcomes have been reported. Yamada et al. and Iuliano et al. have reported that rapid recovery of the ELM and EZ was achieved using the ILMP technique [19–22]. They concluded that part of the ILM in the hole could interrupt the approximation of the outer retina, inducing delayed recovery of the ELM and EZ [19,22]. However, Chou et al. have reported that the ILMF technique was associated with faster recovery [18]. They assumed that the lack of disturbance to glial tissue formation by the ILM flap over the hole promoted rapid recovery [18]. These conflicting results may be attributed to the detailed surgical techniques, including conventional or modified ILMF techniques.

Six studies performed meta-analyses of visual outcomes [17–22]. No significant differences in preoperative and postoperative visual acuity were observed between the two surgical techniques. However, sensitivity tests were applied to resolve heterogeneity because the final visual acuity indicated heterogeneity. Chou et al. have reported that the ILMF group achieved rapid visual recovery at 3 and 6 months, despite no difference in final visual acuity [18]. However, its missing data (40% and 20% in the ILMF group and 6% and 8% in the ILMP group at 3 and 6 months, respectively) limited our understanding [18]. Other studies have not reported any differences in the recovery time of visual acuity between the two techniques.

Two studies were included in this meta-analysis of retinal sensitivity. Contrary outcomes have been reported in previous studies [19,21]. Ventre et al. have reported improved retinal sensitivity in the ILMP group at 6 and 12 months [21]. They measured retinal sensitivity in the range of 8˚. However, Iuiano et al. have demonstrated that the ILMP technique achieved better retinal sensitivity in the 4˚ range but did not demonstrate a difference in the 10˚ range. The meta-analysis of the two studies revealed no differences in the baseline and follow-up visits between the two surgical techniques. Based on these findings, retinal sensitivity may depend on the assessed macular area. In addition, instrumental differences may have affected the outcomes. More studies are needed to compare retinal sensitivity between the ILMF and ILMP techniques.

The present study has several limitations. First, only few studies were included. The sample size of each study was small. Second, the different follow-up periods of the studies could have influenced the surgical and visual outcomes in this analysis because the data were not sufficient to demonstrate complete recovery of visual acuity and outer retinal layers in some studies. Third, because a small number of RCTs were included despite the low bias of the included literature, data quality and selection bias could be problematic factors in the present study. Fourth, bias may be present owing to missing data. Fifth, many other factors that could affect the outcomes, such as the duration of disease, vital dyes, tamponade material, and face-down period, were not considered. Sixth, cataract surgery was performed as required in most studies. Cataract status may also influence visual outcomes. Seventh, the P value of the difference of the two techniques in MH closure rate was 0.09, which tends to be close to a significant difference. The retrospective studies may have an inherent selection bias. About half of eyes were derived from the data of Baumann and colleagues' study (the closure rates of the ILM flap and ILM peeling technique were 100% and 96%, respectively). These might impact on the overall results of our analysis. To clarify this, when excluding the data of Baumann et al., additional

meta-analysis showed that the closure rate of MH approached to 99% (109/110) in the inverted flap technique and 97% (111/115) in the ILM peeling technique, respectively (P = 0.45). This also showed no difference of two surgical techniques in the primary hole closure rate. However, different outcomes may come out if more data can be achieved in future RCT studies. Eighth, the surgeon's factor of the inverted ILM flap technique can be more various than the ILM peeing technique and it may influence the outcomes. Lastly, in the retinal sensitivity analysis, we could not analyze sufficient data from the studies using a uniform assessment.

## Conclusions

No significant differences in hole closure rate, visual prognosis, retinal sensitivity, and outer retinal recovery were observed between the ILMF and ILMP techniques for the treatment of MHs ≤400 μm. Both techniques demonstrated high success rates for hole closure and improved the visual function and outer retinal anatomy. Based on these findings, both surgical techniques can be beneficially used depending on the surgeon's preference for small-to-medium-sized MH. For a more accurate comparison of surgical technique studies, additional RCTs with larger sample sizes and standardized surgical techniques are necessary.

## Supporting information

**S1 Checklist. PRISMA 2009 checklist.**
(DOCX)

## Author Contributions

**Conceptualization:** Seung Min Lee, Iksoo Byon.

**Data curation:** Seung Min Lee.

**Formal analysis:** Seung Min Lee, Iksoo Byon.

**Methodology:** Seung Min Lee, Ji Woong Lee, Ji Eun Lee, Hee-young Choi, Jong Soo Lee, Iksoo Byon.

**Supervision:** Iksoo Byon.

**Validation:** Ji Woong Lee, Ji Eun Lee, Hee-young Choi, Jong Soo Lee.

**Visualization:** Seung Min Lee.

**Writing – original draft:** Seung Min Lee.

**Writing – review & editing:** Ji Woong Lee, Ji Eun Lee, Hee-young Choi, Jong Soo Lee, Iksoo Byon.

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
