## [Decision Letter · Decision Letter 0]

14 Jan 2024

PONE-D-23-38847

Efficacy of inverted inner limiting membrane flap technique for macular holes of ≤400 µm: A systematic review and meta-analysis

PLOS ONE

Dear Dr. Byon,

Thank you for submitting your manuscript to PLOS ONE. After careful consideration, we have decided that your manuscript does not meet our criteria for publication and must therefore be rejected.

I am sorry that we cannot be more positive on this occasion, but hope that you appreciate the reasons for this decision.

Kind regards,

Jiro Kogo

Academic Editor

PLOS ONE

Reviewers' comments:

Reviewer's Responses to Questions

**Comments to the Author**

1. Is the manuscript technically sound, and do the data support the conclusions?

Reviewer #1: Partly

Reviewer #2: Yes

2. Has the statistical analysis been performed appropriately and rigorously? 

Reviewer #1: Yes

Reviewer #2: Yes

3. Have the authors made all data underlying the findings in their manuscript fully available?

Reviewer #1: Yes

Reviewer #2: Yes

4. Is the manuscript presented in an intelligible fashion and written in standard English?

Reviewer #1: Yes

Reviewer #2: Yes

5. Review Comments to the Author

Reviewer #1: I read the article “Efficacy of inverted inner limiting membrane flap technique for macular holes of ≤400 μm: A systematic review and meta-analysis” with interest. The article reviewed articles focusing on small and medium-sized macular holes and attempted to answer the question of whether an inverted internal limiting membrane flap is beneficial for these smaller holes. The authors concluded, after pooled analysis of four retrospective studies and two prospective studies, that either the ILM flap or the ILM peeling achieved a high anatomical success rate, and there were differences between the two. Several comments as below:

1. The main concern is whether the current evidence is powered enough to detect the minor difference, if any, in primary hole closure between ILM flap and ILM peeling. The retrospective studies have an inherent selection bias, and the two enrolled prospective randomized trials consisted of only 66 patients in total. The fact that the calculated P value between the ILM flap and ILM peeling technique on the primary hole closure rate was 0.09, which is approaching 0.05, is alarming.

2. If the current evidence is not sufficient, the conclusion should be stated as such or should state the limitations. The abstract should be revised accordingly.

3. There are different types of macular holes. I presume this study focused on full-thickness macular holes. This should be defined clearly.

4. The title could be revised to show the aim of the study, which was a comparison between ILM peeling and ILM flap.

5. “the primary outcome measure was anatomical hole closure rate” should clearly state whether it was the primary or the secondary hole closure.

6. The ILM flap technique has some variations and potentially affects the outcomes. This should be acknowledged in the discussion or as a limitation of this study.

Reviewer #2: Interesting and well written manuscript. I consider this paper fully suitable for publication on this journal. The topic is interesting. The results are clear. The English language is fluent. No further comments for the authors.

6. PLOS authors have the option to publish the peer review history of their article (what does this mean?). If published, this will include your full peer review and any attached files.

Reviewer #1: No

Reviewer #2: No

- - - - -

---

## [Author Response · Author response to Decision Letter 0]

10 Feb 2024

Dear Editor & Reviewers,

We appreciate all suggestions and corrections. All authors have carefully reviewed these comments. Our detailed responses to comments are addressed below.

Comments to the Author

Reviewer #1: I read the article “Efficacy of inverted inner limiting membrane flap technique for macular holes of ≤400 μm: A systematic review and meta-analysis” with interest. The article reviewed articles focusing on small and medium-sized macular holes and attempted to answer the question of whether an inverted internal limiting membrane flap is beneficial for these smaller holes. The authors concluded, after pooled analysis of four retrospective studies and two prospective studies, that either the ILM flap or the ILM peeling achieved a high anatomical success rate, and there were differences between the two. Several comments as below:

1. The main concern is whether the current evidence is powered enough to detect the minor difference, if any, in primary hole closure between ILM flap and ILM peeling. The retrospective studies have an inherent selection bias, and the two enrolled prospective randomized trials consisted of only 66 patients in total. The fact that the calculated P value between the ILM flap and ILM peeling technique on the primary hole closure rate was 0.09, which is approaching 0.05, is alarming.

2. If the current evidence is not sufficient, the conclusion should be stated as such or should state the limitations. The abstract should be revised accordingly.

Response 1-2: Among studies in this meta-analysis, about half of eyes were derived from the data of Baumann and colleagues’ study (the closure rates of the ILM flap and ILM peeling technique were 100% and 96%, respectively). It might impact on the overall results of our analysis. Therefore, we had excluded the data of Baumann et al. and analyzed two surgical techniques to clarify this. Additional data analysis showed that the closure rate of MH approached to 99% (109/110) in the inverted flap technique and 97% (111/115) in the ILM peeling technique, respectively (P = 0.45). This also showed no difference of two surgical techniques in the primary hole closure rate. As the reviewer pointed out, careful interpretation should be needed in comparison between the ILM flap and ILM peeling technique. The following phrase was added to limitation. 

“Seventh, the P value of the difference of the two techniques in MH closure rate was 0.09, which tends to be close to a significant difference. The retrospective studies may have an inherent selection bias. About half of eyes were derived from the data of Baumann and colleagues’ study (the closure rates of the ILM flap and ILM peeling technique were 100% and 96%, respectively). These might impact on the overall results of our analysis. To clarify this, when excluding the data of Baumann et al., additional meta-analysis showed that the closure rate of MH approached to 99% (109/110) in the inverted flap technique and 97% (111/115) in the ILM peeling technique, respectively (P = 0.45). This also showed no difference of two surgical techniques in the primary hole closure rate. However, different outcomes may come out if more data can be achieved in future RCT studies” (in the page 20)

3. There are different types of macular holes. I presume this study focused on full-thickness macular holes. This should be defined clearly.

Response 3: The term first mentioned in the abstract was changed to “full-thickness macular hole”, and the inclusion criteria (page 4) of materials and methods was changed to “First, for population (participants), patients diagnosed with idiopathic full-thickness MH.”

4. The title could be revised to show the aim of the study, which was a comparison between ILM peeling and ILM flap.

Response 4: We don’t think that the title needs to be changed, regarding of word count (150) and aim of study for the efficacy of the inverted ILM flap technique in small FTMH.

5. “the primary outcome measure was anatomical hole closure rate” should clearly state whether it was the primary or the secondary hole closure.

Response 5: As the reviewer pointed out, the criteria for anatomical closure are specified in the following statement: “Third, for outcome measures, the primary outcome measure was anatomical hole closure rate in the first operation.” (in page 5, second paragraph)

6. The ILM flap technique has some variations and potentially affects the outcomes. This should be acknowledged in the discussion or as a limitation of this study.

Response 6: As the reviewer’s comments, the following phrases were added in the limitations: “Eighth, the surgeon’s factor of the inverted ILM flap technique can be more various than the ILM peeing technique and it may influence the outcomes.” (in the page 21)

Reviewer #2: Interesting and well written manuscript. I consider this paper fully suitable for publication on this journal. The topic is interesting. The results are clear. The English language is fluent. No further comments for the authors.

Response: Thank you.

---

## [Decision Letter · Decision Letter 1]

5 Apr 2024

Efficacy of inverted inner limiting membrane flap technique for macular holes of ≤400 µm: A systematic review and meta-analysis

PONE-D-23-38847R1

Dear Dr. Byon,

We’re pleased to inform you that your manuscript has been judged scientifically suitable for publication and will be formally accepted for publication once it meets all outstanding technical requirements.

Kind regards,

Koichi Nishitsuka

Academic Editor

PLOS ONE

Additional Editor Comments (optional):

Reviewers' comments:

Reviewer's Responses to Questions

**Comments to the Author**

1. If the authors have adequately addressed your comments raised in a previous round of review and you feel that this manuscript is now acceptable for publication, you may indicate that here to bypass the “Comments to the Author” section, enter your conflict of interest statement in the “Confidential to Editor” section, and submit your "Accept" recommendation.

Reviewer #1: All comments have been addressed

Reviewer #2: All comments have been addressed

2. Is the manuscript technically sound, and do the data support the conclusions?

Reviewer #1: Yes

Reviewer #2: Yes

3. Has the statistical analysis been performed appropriately and rigorously? 

Reviewer #1: Yes

Reviewer #2: Yes

4. Have the authors made all data underlying the findings in their manuscript fully available?

Reviewer #1: Yes

Reviewer #2: Yes

5. Is the manuscript presented in an intelligible fashion and written in standard English?

Reviewer #1: Yes

Reviewer #2: (No Response)

6. Review Comments to the Author

Reviewer #1: I thank the authors for revising the manuscript. All my concerns have been addressed. I have no further comments for this work.

Reviewer #2: The authors well addressed all the comments. No further comments. I encourage the publication of the manuscript

7. PLOS authors have the option to publish the peer review history of their article (what does this mean?). If published, this will include your full peer review and any attached files.

Reviewer #1: No

Reviewer #2: No
